# Thermodynamic Analysis and Optimization of a Novel Power-Water Cogeneration System for Waste Heat Recovery of Gas Turbine

**DOI:** 10.3390/e23121656

**Published:** 2021-12-09

**Authors:** Shunsen Wang, Bo Li

**Affiliations:** State Key Laboratory of Multiphase Flow in Power Engineering, Institute of Turbomachinery, School of Energy and Power Engineering, Xi’an Jiaotong University, No. 28 Xianning West Road, Xi’an 710049, China; xijiaodalibo@stu.xjtu.edu.cn

**Keywords:** power-water cogeneration system, supercritical carbon dioxide Brayton cycle, reverse osmosis, thermodynamic analysis, optimization

## Abstract

A power-water cogeneration system based on a supercritical carbon dioxide Brayton cycle (SCBC) and reverse osmosis (RO) unit is proposed and analyzed in this paper to recover the waste heat of a gas turbine. In order to improve the system performance, the power generated by SCBC is used to drive the RO unit and the waste heat of SCBC is used to preheat the feed seawater of the RO unit. In particular, a dual-stage cooler is employed to elevate the preheating temperature as much as possible. The proposed system is simulated and discussed based on the detailed thermodynamic models. According to the results of parametric analysis, the exergy efficiency of SCBC first increases and then decreases as the turbine inlet temperature and split ratio increase. The performance of the RO unit is improved as the preheating temperature rises. Finally, an optimal exergy efficiency of 52.88% can be achieved according to the single-objective optimization results.

## 1. Introduction

In order to deal with the severe challenges of excessive carbon emission and global warming, it is necessary to develop an efficient and clean power generation system. In this context, gas turbine fueled by natural gas would play an important role in the future energy market [1,2]. Although gas turbines have been widely used as prime engines in various industrial fields like power plants and transportation sectors, the thermal efficiency of a standalone gas turbine is still limited since the majority of heat input is discharged into the environment as waste heat [3]. Therefore, the waste heat recovery system has become a research hotspot in the past few years and lots of relevant work is being conducted to improve the efficiency of gas turbine [4,5].

Owing to its features of structural compactness, high efficiency, and nontoxicity, the supercritical carbon dioxide Brayton cycle (SCBC) has shown significant advantages integrated with concentrated solar power [6], nuclear power [7], and a coal-fired power plant [8] in terms of steady thermal performance and off-design performance. Recently, the supercritical carbon dioxide power cycle was further introduced to recover the waste heat of a gas turbine by many researchers and companies. Cao et al. [9] analyzed the thermodynamic performance of a cascade carbon dioxide cycle driven by the waste heat of a gas turbine. Their results showed that the thermal efficiency of the investigated cascade system is 17.03% higher than that of an original gas turbine. Zhang et al. [10] proposed a novel cascade carbon dioxide cycle to recover the waste heat in which the CO_2_ is pre-compressed by a pump to fully exploit the advantages of carbon dioxide thermodynamic properties. They concluded that the net power output was enhanced by 5.3% compared with the typical layout. Budiyanto et al. [11] developed a cascade CO_2_ cycle to recover the waste heat of a gas turbine as well as the cold energy of liquefied natural gas. They indicated that the optimal exergy efficiency of the proposed system was up to 60.93%. Song et al. [12] established and evaluated a modified preheating supercritical CO_2_ cycle to utilize the residual heat of exhausted gas and jacket cooling water. They reported that the net power output was elevated by 7.4%. Villafana and Bueno [13] investigated the performance of partial heating SCBC for the waste heat recovery of the air Brayton cycle via thermo-economic and environmental analysis. They pointed out that the total cost of the combined system was reduced by 10.37%. Li et al. [14] evaluated the off-design performance of a partial heating cycle under variable flue gas parameters and ambient temperature. The corresponding control strategies were developed to maximize the system exergy efficiency under different operating conditions. The abovementioned carbon dioxide cycles are proposed and studied under different boundary conditions. Therefore, it is necessary to compare their performance under the same framework. Li et al. [15] compared six typical supercritical carbon dioxide Brayton cycles designed for high-temperature waste heat recovery. Generally speaking, the partial heating cycle was recommended because of its balanced overall performance. The partial heating cycle was also recommended by Kim et al. [16]. Therefore, the partial heating carbon dioxide cycle is employed to recover the waste heat and generate power in this paper.

With the development of economy and society, the demand for fresh water is increasing rapidly. There are many mature technologies for seawater desalination: multiple effect distillation, multi-stage flash, reverse osmosis (RO), solar still, and so on [17]. Among all these desalination technologies, RO is the most widely used process due to its simple layout as well as high efficiency. Nearly 65% of the fresh water around the world is produced by a RO system from seawater [18]. RO desalination process is typically a high energy consumption process, requiring approximately 2.5–5 kWh of power to produce 1 m^3^ fresh water even with energy recovery devices [19]. Therefore, the combined power and water cogeneration system has been studied by many researchers. Eveloy et al. [20] proposed and analyzed a combined organic Rankine cycle and RO system driven by the waste heat of gas turbine. They claimed that the power generation efficiency was improved by approximately 12%. Geng et al. [21] analyzed the coupled RO unit and organic Rankine cycle system with zeotropic mixtures as working fluid. They found that the utilization of zeotropic mixtures could enhance the performance of the cogeneration system due to their temperature glide characteristics in the phase transition process. Altmann et al. [22] analyzed and compared a series of power-water cogeneration systems from the perspective of energy and exergy. Musharavati et al. [23] proposed a poly-generation system in which the Kalina cycle was utilized to drive a RO subsystem. Jafarzad et al. [24] carried out thermodynamic analysis on a combined organic Rankine cycle and RO system for the waste heat recovery of diesel engine. They found the maximum exergy efficiency of the cogeneration system was 54.10%.

The power-water cogeneration system consisting of a carbon dioxide power cycle and reverse osmosis unit has been preliminarily studied in some previous literature. Xia et al. [25] proposed a RO desalination process powered by a solar-driven transcritical CO_2_ power cycle with liquefied natural gas as a heat sink. Their results showed that turbine inlet pressure had a great impact on the system performance and an optimal exergy efficiency of 4.9% could be achieved under the design conditions. Naseri et al. [26] modified Xia’s work [25] by adding a hydrogen production function and an auxiliary boiler. They found that the solar collector and condenser dominated the exergy destruction and there was an optimal value of turbine inlet pressure to maximize the system output. Manesh et al. [27] evaluated a waste heat recovery system composed of a supercritical carbon dioxide cycle, organic Rankine cycle, and reverse osmosis desalination unit. They indicated that the proposed system could achieve high thermal efficiency and low economic cost.

According to the literature review, it could be concluded that the research on waste heat recovery of gas turbine and power-water cogeneration systems are important and necessary. However, to the best of our knowledge, studies on a combined supercritical carbon dioxide cycle and RO unit driven by the waste heat of gas turbine is lacking. In the previous papers, the power cycle and RO unit are only mechanically connected, which means the RO unit is driven by the power cycle. It is worth noting that the performance of the RO unit would improve as the operating temperature increases [28]. The low-temperature waste heat is a perfect heat source to preheat the feed seawater of RO unit. Therefore, the power cycle and RO unit should be connected both mechanically and thermally to further improve the system efficiency. Overall, a power-water cogeneration system composed of a supercritical carbon dioxide power cycle and RO unit is proposed to recover the waste heat of a gas turbine. The power generated by the carbon dioxide cycle is used to drive the RO unit and the low-temperature waste heat is used to preheat the feed seawater of the RO unit. The system configuration and mathematical models are described in detail. The parametric analysis and optimization are conducted to investigate the performance of the proposed system.

## 2. System Configuration

The schematic diagram of the proposed power-water cogeneration system consisting of a supercritical carbon dioxide Brayton cycle and reverse osmosis unit is shown in Figure 1. The high-temperature high-pressure carbon dioxide (Point 1) expands in the turbine (Tur) to generate power. Then, the exhausted working fluid (Point 2) flows into the recuperator (Rec) and cooler successively to release heat. After that, the carbon dioxide (Point 4) is compressed to high pressure by the compressor (Com). The compressed working fluid (Point 5) is divided into two parts: the first part of carbon dioxide flows into the low-temperature heater (LTH) to absorb heat from flue gas; while the other part enters the recuperator to absorb heat from the low-pressure carbon dioxide. The working fluid mixes at Point 6 and then absorbs heat in the high-temperature heater (HTH) to complete the supercritical carbon dioxide Brayton cycle. 

As for the reverse osmosis unit, the seawater (Point 01) is preheated by the flue gas and the waste heat of SCBC, respectively. The preheated feed seawater (Point 03) is pumped to high pressure by a high-pressure pump (HPP) and separated by reverse osmosis. The permeate water (Point 05) is collected into a tank and the brine water (Point 06) is discharged after passing through an energy recovery turbine (ERT). 

In order to simplify the simulation, several common assumptions are adopted as follows:(1)The proposed cogeneration system operates in steady state.(2)Pressure losses and heat losses of the pipes are ignored.(3)The heat loss of heat exchangers is neglected. The pressure drop of the heat exchangers is assumed as 1% of inlet pressure.(4)The carbon dioxide is mixed with the same thermodynamic properties at Point 6.

## 3. Mathematical Modeling

In this section, the thermodynamic and economic models of the components in the proposed power-water cogeneration system are established. Then, some performance criterions are proposed to evaluate the system performance from the different perspectives. Finally, model validation is conducted to prove the correctness of the models.

### 3.1. Modeling of SCBC

The thermodynamic models of the supercritical carbon dioxide power cycle are established based on the first and second law of thermodynamics. The general equations for mass balance, energy balance, and exergy balance can be expressed as [15]:(1)$$\sum {\dot{m}}_{in}=\sum {\dot{m}}_{out}$$
(2)$$\sum {\left(\dot{m}h\right)}_{in}-\sum {\left(\dot{m}h\right)}_{out}+\sum {\dot{Q}}_{in}-\sum {\dot{Q}}_{out}+\dot{W}=0$$
(3)$${\dot{E}}_{Q}+\sum {\left(\dot{m}e\right)}_{in}={\dot{E}}_{W}+\sum {\left(\dot{m}e\right)}_{out}+{\dot{E}}_{des}$$
where $\dot{m}$, $\dot{Q}$, $\dot{E}$, $\dot{W}$ represent mass flow rate, heat flow rate, exergy flow rate, and power rate, respectively; *h* and *e* mean specific enthalpy and specific exergy; in and out denote inlet and outlet.

Putting the corresponding items into the Equations (1)–(3), the thermodynamic models of SCBC components can be obtained. Since the models of carbon dioxide power cycle have been discussed sufficiently in the previous literatures, they will not be discussed in detail herein. The model equations of SCBC are summarized in Table 1.

### 3.2. Modeling of RO Unit

The models of reverse osmosis unit are established based on [29,30].

The mass balance equations of seawater and salt can be expressed as:(4)$${\dot{Q}}_{f}={\dot{Q}}_{r}+{\dot{Q}}_{p}$$
(5)$${\dot{Q}}_{f}C_{f}={\dot{Q}}_{r}C_{r}+{\dot{Q}}_{p}C_{p}$$
where $\dot{Q}$ and *C* are the flow rate and concentration; *f*, *r*, *p* means feed seawater, retentate solution, and permeate water.

The water flux ($J_{w}$) and salt flux ($J_{s}$) passing through the reverse osmosis can be calculated by the following equations:(6)$$J_{w}=A_{w}\left(\Delta P-\Delta \pi \right)$$
(7)$$J_{s}=A_{s}\left(C_{m}-C_{p}\right)$$
where $A_{w}$ and $A_{s}$ are the water permeability constant and solute transport constant at operating temperature; ΔP and Δπ are the operating pressure difference and osmotic pressure difference across the membrane; $C_{m}$ is the membrane surface concentration.

The membrane surface concentration is estimated according to film theory:(8)$$\frac{C_{m}-C_{p}}{C_{b}-C_{p}}=\exp\left(J_{w}/k\right)$$
where k is the mass transfer coefficient, which is determined by the operating parameters and structural parameters.

The flow rate and concentration of permeate water could also be expressed as:(9)$${\dot{Q}}_{p}=AJ_{w}$$
(10)$$C_{p}=J_{s}/J_{w}$$

The pressure drop along the membrane element is calculated by:(11)$$\Delta P_{drop}=\frac{\rho u^{2}LC_{td}}{2d_{h}}$$
where *L*, $C_{td}$, and $d_{h}$ are structural parameters of the RO unit.

The power consumed by the RO unit is expressed as:(12)$${\dot{W}}_{RO}={\dot{W}}_{HPP}-{\dot{W}}_{ERT}$$

The performance of the RO unit can be evaluated by the water recovery ratio (WRR) and specific energy consumption (SEC):(13)$$\mathrm{WRR}={\dot{Q}}_{p}/{\dot{Q}}_{f}$$
(14)$$\mathrm{SEC}={\dot{W}}_{RO}/{\dot{Q}}_{p}$$

### 3.3. Economic Model

Economic models of the key components are established as the function of device sizes and design parameters. Economic index is evaluated in order to avoid pursuing thermal efficiency at the expense of unreasonable cost. The equipment in the carbon dioxide power cycle can be classified into a heat exchanger and a turbomachine. The investment of the heat exchanger is assumed as the function of heat conductance (heat transfer coefficient multiplies the heat transfer area) and the investment of the turbomachine is assumed as the function of the power rate.

The investment cost of reverse osmosis is composed of the cost of water pretreatment, membrane module, high-pressure pump, and energy recovery turbine, which are proportional to the mass flow rate of feed seawater. Overall, the investment cost of the proposed system can be evaluated by the equations listed in Table 2.

### 3.4. Evaluation Criteria

Apparently, the power output of the supercritical carbon dioxide cycle (${\dot{W}}_{SCBC}$) and the mass flow rate of permeate water (${\dot{Q}}_{p}$) should be adopted as evaluation criteria of the two subsystems.
(15)$${\dot{W}}_{SCBC}={\dot{W}}_{Tur}-{\dot{W}}_{Com}$$

Further, exergy efficiency is defined for the two subsystems:(16)$$\eta_{SCBC}={\dot{W}}_{SCBC}/\left({\dot{m}}_{g1}e_{g1}\right)$$
(17)$$\eta_{RO}={\dot{W}}_{RO,least}/{\dot{W}}_{RO}$$

${\dot{W}}_{RO,least}$ is the least energy consumption to separate the water and solute when the recovery ratio of RO subsystem is close to zero [32].

The exergy efficiency of the power-water cogeneration system can be expressed as:(18)$$\eta_{system}=\left({\dot{W}}_{SCBC}+{\dot{W}}_{RO,least}\right)/\left({\dot{m}}_{g1}e_{g1}\right)$$

### 3.5. Model Validation

The accessible data in previous literature are utilized to prove the correctness of our models. Because there is no earlier study on the proposed system by experiment or simulation, the validation will be conducted on the subsystems. Table 3 and Table 4 display the comparison results of the CO_2_ cycle subsystem and RO unit between the predicted results and the data from previous literature [16,30], respectively. It is obvious that the deviations of all models are within the acceptable ranges. Therefore, it can be concluded that the established models can evaluate the performance of proposed system accurately.

## 4. Results and Discussion

In this section, the performance of the proposed system is simulated and evaluated. The input parameters under the design conditions are displayed in Table 5 [15,30]. The simulation is carried out in MATLAB software and the thermodynamic properties of the carbon dioxide are obtained from REFPROP 9.1. Some important parameters, such as turbine inlet temperature (T_1_), split ratio (SR), and preheating temperature (T_03_), are analyzed to investigate their effects on the system performance. When one parameter is investigated, other parameters are fixed as design values. All parameters vary in reasonable ranges in the parametric analysis.

### 4.1. The Preheating Configuration

According to our previous paper [33], the residual heat contained in the waste heat of the power cycle and unexploited flue gas are basically in the same magnitude. Therefore, as shown in Figure 1, the feed seawater is divided into two parts and it is preheated in the preheater and cooler, respectively.

It is worth noting that the structure of the cooler is different from typical heat exchangers. The traditional heat exchanger is named as a single-stage cooler while the heat exchanger adopted in this paper is named as a dual-stage cooler. In the dual-stage cooler, the feed seawater firstly flows into the cooler to cool the carbon dioxide. Then, part of the seawater is discharged directly while the other part of the seawater is further preheated to a higher temperature. For comparison, two different types of heat exchangers are shown in Figure 2a and the corresponding temperature-heat load diagram is shown in Figure 2b. The advantages of a dual-stage cooler can be explained by the unique thermodynamic properties of carbon dioxide. When the state point is far away from the critical point, the heat capacity of carbon dioxide is almost constant. However, as the state point gets closer to the critical point, the heat capacity increases dramatically. On the contrary, the heat capacity of cooling water is almost constant. Thus, in the temperature-heat diagram, the slope of the water side is first bigger then smaller than that of the CO_2_ side. If a traditional heat exchanger is adopted as the cooler, the pinch point temperature difference might be very small when the mass flow rate of the cooling water is small, as shown by the blue line in Figure 2b. On the other hand, the outlet temperature of the seawater might not be preheated to the given preheating temperature when the mass flow rate of the cooling water is large, as shown by the green line in Figure 2b. In order to overcome the problem, the dual-stage cooler is employed. By discharging the majority of the feed seawater in the middle of the cooler, the average temperature difference of the cooler can be controlled within an acceptable range and the remaining part of the seawater can be preheated sufficiently, as shown by the red dotted line in Figure 2b.

### 4.2. Case Study

In this section, the exergy analysis and economic analysis are carried out for the design case. As shown in Figure 3, the exergy destruction of the heat exchangers is higher than the turbomachines. The exergy destruction of the cooler and high-temperature heater dominates the exergy destruction of the proposed water-power cogeneration system, which is caused by the numerous heat transfer amount and large heat transfer temperature difference. The exergy destruction related with the turbine is also large because of the irreversibility of the expansion process. The “Others” in Figure 3 mainly involves the unexploited exergy of the flue gas, which is discharged into the environment. It can be seen that the unexploited exergy is very small compared with the total exergy input of the flue gas, which indicates the proposed water-power cogeneration system can recover the waste heat of the gas turbine sufficiently and effectively. Compared with the carbon dioxide power cycle, the exergy destruction of RO unit is relatively small since the water productivity of the RO unit is low. However, if the requirement for fresh water increases, the exergy destruction of the RO unit will rise proportionally.

The investment costs of the components are shown in Figure 4. The cost of the “heater” includes the cost of the high-temperature heater, low-temperature heater, and preheater. Apparently, the investment cost of turbine occupied nearly half of the proposed water-power cogeneration system. On the contrary, the cost of the heater is relatively small because of the large temperature difference. As explained above, the RO unit further utilizes the waste heat of the flue gas and carbon dioxide cycle, which greatly improves the thermodynamic performance of the waste heat recovery system. Meanwhile, the investment cost of the RO unit is only 9.28% of the total cost. Therefore, the utilization of the RO unit would enhance the system thermal performance at a relatively small expense, leading to satisfactory overall performance.

### 4.3. Parametric Analysis

Figure 5 shows the effect of turbine inlet temperature on the performance of proposed power-water cogeneration system. The power output of supercritical carbon dioxide cycle first increases then decreases as the turbine inlet temperature increases. According to the basic knowledge of thermodynamics, the specific work of a turbine will rise with the increment of inlet temperature under a constant pressure ratio. Nevertheless, the temperature of Point 2 and Point 6 also increases, leading to a narrowing temperature range of the high-temperature heater. Meanwhile, as the turbine inlet temperature increases, the heat absorbed by per mass flow rate of carbon dioxide in the high temperature heater shows an ascending trend. Therefore, the mass flow rate of carbon dioxide decreases. Considering the effects mentioned above, the variation tendencies of exergy efficiency and power output of SCBC can be explained. The performance of the reverse osmosis unit can only be affected by its operating parameters, such as the preheating temperature, the mass flow rate of feed seawater, etc. Since these parameters are constant, the performance of the RO unit will not change with the turbine inlet temperature and are not displayed in Figure 5.

Figure 6 illustrates the effect of a split ratio on the system performance of the proposed cogeneration system. The splitting configuration is employed owing to the unique thermodynamic properties of carbon dioxide. When the operating temperature is near critical point, the specific heat capacity of carbon dioxide with high pressure is much higher than that of carbon dioxide with low pressure. Therefore, part of carbon dioxide is delivered into a low-temperature heater to make the heat capacity of the recuperator matched. Obviously, there is an optimal value of the split ratio at which the heat capacities of both sides of the recuperator are exactly matched. If the split ratio is larger than the optimal value, the carbon dioxide on the cold side of the recuperator cannot be preheated sufficiently. If the split ratio is smaller than the optimal value, the residual heat of carbon dioxide on the hot side of the recuperator cannot be fully recovered. Thereby, the power output and exergy efficiency of SCBC first increases then decreases with the increment of split ratio. Because of the same reason mentioned above, the performance of the RO unit will not be affected by the variation of the split ratio.

Figure 7 displays the effect of feed seawater preheating temperature on the system performance of the RO unit. As the preheating temperature increases, the water recovery ratio of the RO unit increases and the specific energy consumption of the RO unit decreases. The membrane transport parameters of water and solute increases with the increment of operating temperature [28]. Since the other parameters are constant, the mass flow rate of water passing through the membrane should show a rising trend, leading to a higher water recovery ratio. However, the solute passing through the membrane would also increase, which means the concentration of permeate water rises. On the other hand, the power consumption of RO unit is basically decided by the high-pressure pump and energy recovery turbine. Therefore, the specific energy consumption declines since the mass flow rate of permeate water increases.

### 4.4. Optimization

According to the previous parametric analysis, the system performance would be greatly affected by the variation of design parameters. Apparently, the system efficiency might be maximized at an optimal combination of design variables. In order to find the optimal design parameters, single-objective optimization is carried out and a genetic algorithm is employed as an optimization method. A genetic algorithm is a self-adaptive global optimization algorithm inspired by the evolution process of living beings in the natural environment. A population of suitable size is initialized and all individuals are evaluated according to the function. The individuals most adapted to the objective function together with the new individuals generated by crossover and mutation processes form a new population. This process is be repeated until the converge condition is achieved. Finally, the optimal solution is obtained.

Based on the parametric analysis, the turbine inlet temperature (T_1_), split ratio (SR), and preheating temperature (T_03_) are adopted as design variables. Additionally, the system exergy efficiency is employed as the optimization objective. The mass flow rate of permeate water is fixed at 50 kg/s in this section. The single-objective optimization is carried out based on the optimization toolbox built in MATLAB platform. The boundary conditions and settings for the single-objective optimization are listed in Table 6 and the optimized results are given in Table 7.

According to Table 4, the optimal turbine inlet temperature and split ratio are 641.18K and 0.672, respectively, at which the exergy efficiency of the supercritical carbon dioxide cycle is up to 55.08%. The optimal preheating temperature is equal to 318.15K and the exergy efficiency of RO unit reaches 27.99%. Overall, the maximum exergy efficiency of the proposed power-water cogeneration system for waste heat recovery is 52.88%.

## 5. Conclusions

In this paper, a power and water cogeneration system is proposed to recover the waste heat of a gas turbine. Detailed thermodynamic and economic models of the components are established to simulate the system. Parametric analysis and optimization are carried out to investigate the system performance. The main conclusions and contributions of this work are summarized below:(1)The feed seawater is preheated by the waste heat of the carbon dioxide cycle and low-temperature flue gas simultaneously. Additionally, a dual-stage cooler is employed to elevate the preheating temperature as much as possible.(2)The exergy efficiency of the supercritical carbon dioxide cycle first increases and then decreases with the increment of turbine inlet temperature and split ratio. The system performance of the RO unit would be improved as the preheating temperature increases.(3)The exergy efficiency of the proposed power-water cogeneration system is optimized based on a genetic algorithm. According to the optimization results, a maximum exergy efficiency of 52.88% can be achieved, which indicates the system performance is satisfactory.

## Figures and Tables

**Figure 1 entropy-23-01656-f001:**
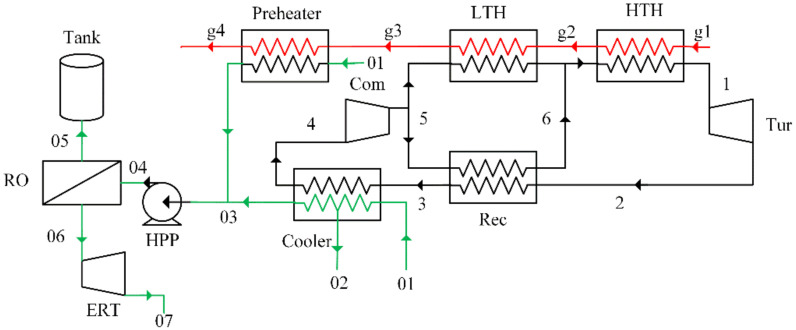
The schematic diagram of the proposed power-water cogeneration system.

**Figure 2 entropy-23-01656-f002:**
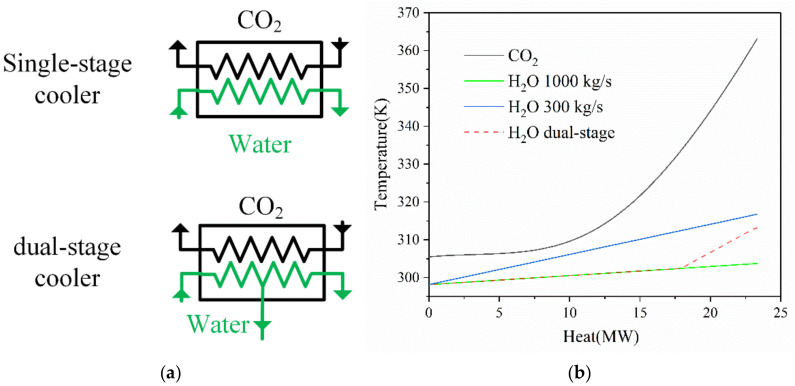
(**a**) The schematic diagram of the single-stage and dual-stage cooler; (**b**) Temperature—heat load diagram of the cooler.

**Figure 3 entropy-23-01656-f003:**
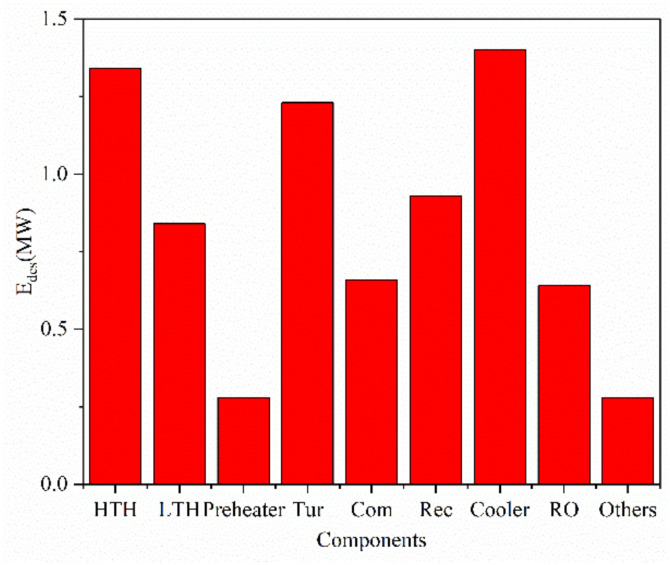
The exergy destruction of components in design case.

**Figure 4 entropy-23-01656-f004:**
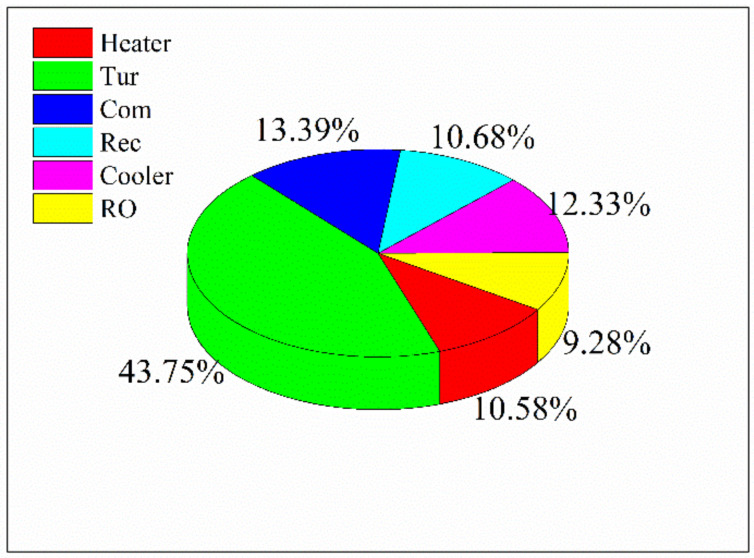
The investment cost of the components in design case.

**Figure 5 entropy-23-01656-f005:**
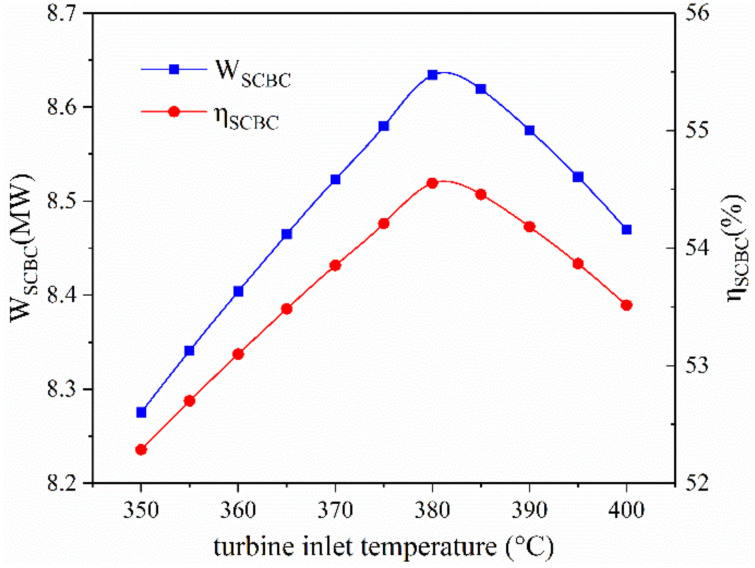
The effect of the turbine inlet temperature on system performance.

**Figure 6 entropy-23-01656-f006:**
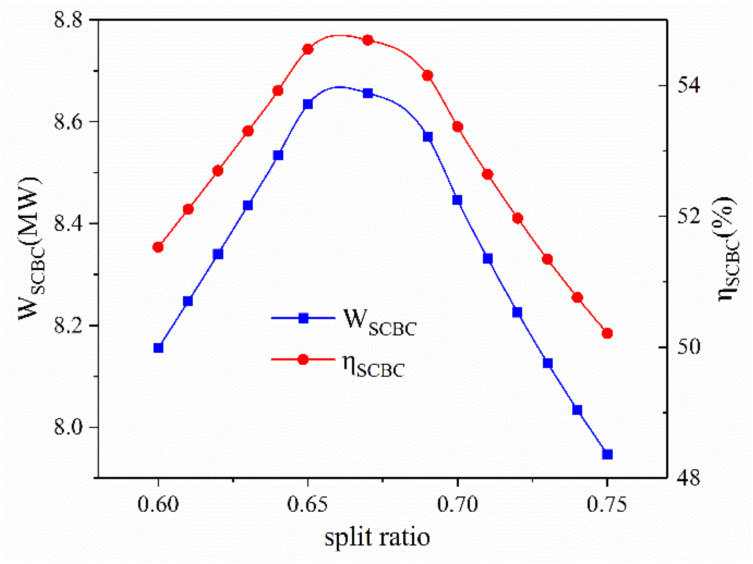
The effect of split ratio on system performance.

**Figure 7 entropy-23-01656-f007:**
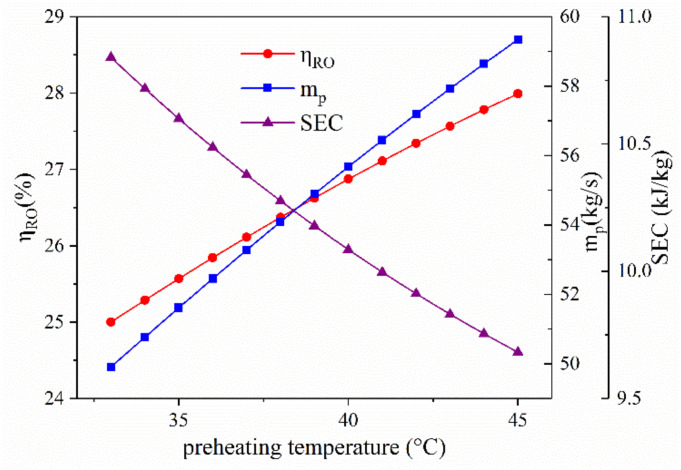
The effect of the preheating temperature on system performance.

**Table 1 entropy-23-01656-t001:** Thermodynamic models of the components in the SCBC.

Components	Models
Tur	${\dot{W}}_{\mathrm{Tur}}={\dot{m}}_{1}\left(h_{1}-h_{2}\right)={\dot{m}}_{1}\left(h_{1}-h_{2s}\right)\eta_{\mathrm{tur}}$
Rec	${\dot{m}}_{1}\left(h_{2}-h_{3}\right)=SR\times {\dot{m}}_{1}\left(h_{6}-h_{5}\right)$
Com	${\dot{W}}_{\mathrm{com}}={\dot{m}}_{1}\left(h_{5}-h_{4}\right)={\dot{m}}_{1}\left(h_{5s}-h_{4}\right)/\eta_{\mathrm{com}}$
HTH	${\dot{m}}_{g1}\left(h_{g1}-h_{g2}\right)={\dot{m}}_{1}\left(h_{1}-h_{6}\right)$
LTH	${\dot{m}}_{g1}\left(h_{g2}-h_{g3}\right)=\left(1-SR\right){\dot{m}}_{1}\left(h_{6}-h_{5}\right)$
Cooler	${\dot{Q}}_{\mathrm{cooler}}={\dot{m}}_{1}\left(h_{3}-h_{4}\right)$

**Table 2 entropy-23-01656-t002:** Cost functions of components in the proposed system [15,31].

Components	Cost Functions
Tur	1000$/kW
Com	1000$/kW
Heater	5000$/(kW/K)
Recuperator	2500$/(kW/K)
Cooler	1700$/(kW/K)
Water pretreatment	996($86400Q_{f}{}^{0.8}$)
HPP	52${\left(3600Q_{f}\left(0.101P_{f}\right)\right)}^{0.96}$
RO membrane	1000$

**Table 3 entropy-23-01656-t003:** Model validation of the carbon dioxide power cycle.

State Points	Temperature (K)		
	This Paper	Literature [16]	Deviation (%)
1	624.91	624.91	0
2	510.54	510.62	0.02
3	361.13	362.04	0.25
4	310	310	0
5	348.8	348.74	0.02
6	502.44	501.68	0.15

**Table 4 entropy-23-01656-t004:** Model validation of the carbon dioxide power cycle.

Cases	Parameters	Experiments [30]	Simulations	Deviation (%)
1	$Q_{f}$ (m^3^/s)	0.0079	0.0079	
	$P_{f}$ (atm)	50.48	50.48	
	$Q_{r}$ (m^3^/s)	0.0063	0.00634	
	$Q_{p}$ (m^3^/s)	0.0016	0.00156	
	Rec (%)	20	19.75	1.25
2	$Q_{f}$ (m^3^/s)	0.0072	0.0072	
	$P_{f}$ (atm)	50.74	50.74	
	$Q_{r}$ (m^3^/s)	0.0056	0.005628	
	$Q_{p}$ (m^3^/s)	0.0016	0.001572	
	Rec (%)	22	21.83	0.77
3	$Q_{f}$ (m^3^/s)	0.0066	0.0066	
	$P_{f}$ (atm)	51.1	51.1	
	$Q_{r}$ (m^3^/s)	0.0050	0.00502	
	$Q_{p}$ (m^3^/s)	0.0016	0.00158	
	Rec (%)	24	23.94	0.25
4	$Q_{f}$ (m^3^/s)	0.0056	0.0056	
	$P_{f}$ (atm)	52.05	52.05	
	$Q_{r}$ (m^3^/s)	0.0040	0.00401	
	$Q_{p}$ (m^3^/s)	0.0016	0.00159	
	Rec (%)	28	28.39	1.39

**Table 5 entropy-23-01656-t005:** Design parameters of the proposed power-water cogeneration system.

Parameters	Values
T_0_	298.15 K
P_0_	0.1013 MPa
T_1_	653.15 K
P_5_	25 MPa
P_4_	7.63 MPa
SR	0.65
$\eta_{\mathrm{Tur}}$	0.85
$\eta_{\mathrm{Com}}$	0.8
$\eta_{\mathrm{HHP}}$	0.8
$\eta_{\mathrm{ERT}}$	0.8
Membrane type	TM820M-400/SWRO
T_03_	313.15 K

**Table 6 entropy-23-01656-t006:** Boundary conditions and settings for the single-objective optimization.

Items	Values
Population size	150
Turbine inlet temperature (K)	623.15–693.15
Split ratio	0.5–0.8
Preheating temperature (K)	305.15–318.15

**Table 7 entropy-23-01656-t007:** Optimization results of the proposed system.

Items	Values
Turbine inlet temperature (K)	641.18
Split ratio	0.672
Preheating temperature (K)	318.15
$\eta_{System}$ (%)	52.88
$\eta_{SCBC}$ (%)	55.08
$\eta_{RO}$ (%)	27.99

## Data Availability

The data is contained within the article.

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
