# Peer review of "Thermodynamic Analysis and Optimization of a Novel Power-Water Cogeneration System for Waste Heat Recovery of Gas Turbine"

_entropy, 2021, doi:10.3390/e23121656_

Round 1

Reviewer 1 Report

The article is interesting and is worth to be published, but not in Entropy. I reccommend to be transfered to an Applied Engineering journal.

Author Response

We would like to thank the dear reviewer for spending time to read our manuscript.

Reviewer 2 Report

Suggest minor corrections to the use of English language, such as "It's" should be "It is" in line 94 and "approximately" instead of "about" on line 66. Additially correct the use of powers, e.g. "1m3" on line 62.

Introduction

Overall good structure to the introduction showing good justification for the decisions made for the paper.

Consider involving more detail regarding some of the discussed studies. For example, Zhang et al proposed a novel design compared to a typical design. What was the novel design? and what classes as a typical design?

Minor corrections to grammar such as:
-Line 52: a sentence should not begin with “And”
-Line 62: “1m3” needs a power on the “3”
-Line 66: the word “about” is informal, perhaps use approximately instead

System Configuration

Suggest adjusting the numbers in Fig. 1 as there are multiple “01” which adds confusion. Above the figure when describing the process refer to all of these numbers to add visual reference for the flow path of the different cycle.

Results and Discussion

More detail needed for the single-objective optimization method. Specifically how are the optimization results generated using the optimization toolbox from MATLAB.

Reviewer 3 Report

The authors proposed and analyzed a power-water cogeneration system to recover the waste heat of the gas turbine. They also employed a dual-stage cooler to increase the preheating temperature. Finally, they used a genetic evaluation algorithm to optimize the exergy energy. 

The authors discussed the details of the previous studies on supercritical carbon dioxide Brayton cycles developed for waste heat recovery. They also found that the combined supercritical carbon dioxide Brayton cycles and reverse osmosis unit driven by the waste heat of gas turbine is not developed yet. So, they took this opportunity to establish a combined model. 

The core heat and mass transfer balance equations are developed in detail and discussed in the paper. The authors also discussed why the use of a dual-stage cooler is advantageous to the regular unit. Finally, graphical representations of the results are easy to read and understand.

Author Response

Thanks for your time to read our paper and your appreciation encourages us to do better work in the future.